# Visual Parking Occupancy Detection Using Extended Contextual Image Information via a Multi-Branch Output ConvNeXt Network

**DOI:** 10.3390/s23063329

**Published:** 2023-03-22

**Authors:** Leyre Encío, César Díaz, Carlos R. del-Blanco, Fernando Jaureguizar, Narciso García

**Affiliations:** Grupo de Tratamiento de Imágenes (GTI), Information Processing and Telecommunications Center, Universidad Politécnica de Madrid, 28040 Madrid, Spain

**Keywords:** parking, detection, parking lot, convolutional neural networks, ConvNeXt, deep learning, computer vision

## Abstract

Along with society’s development, transportation has become a key factor in human daily life, increasing the number of vehicles on the streets. Consequently, the task of finding free parking slots in metropolitan areas can be dramatically challenging, increasing the chance of getting involved in an accident and the carbon footprint, and negatively affecting the driver’s health. Therefore, technological resources to deal with parking management and real-time monitoring have become key players in this scenario to speed up the parking process in urban areas. This work proposes a new computer-vision-based system that detects vacant parking spaces in challenging situations using color imagery processed by a novel deep-learning algorithm. This is based on a multi-branch output neural network that maximizes the contextual image information to infer the occupancy of every parking space. Every output infers the occupancy of a specific parking slot using all the input image information, unlike existing approaches, which only use a neighborhood around every slot. This allows it to be very robust to changing illumination conditions, different camera perspectives, and mutual occlusions between parked cars. An extensive evaluation has been performed using several public datasets, proving that the proposed system outperforms existing approaches.

## 1. Introduction

Along with society’s development, transportation has become a key factor in human daily life, leading to an exponential increase in the number of vehicles on the streets. As reported by the Spanish Directorate-General for Roads, a department of the Ministry of Transport, Mobility and Urban Agenda, the 2021 Annual average daily traffic (AADT) of Madrid’s main roads was over 23,000 vehicles per road section and day [1]. In addition, the AADT has undergone an increment of 10% between 2012 and 2021 [1]. This fact especially affects the chance of finding free parking spaces. According to several studies, an average of approximately 30% of vehicles on the road in metropolitan areas are searching for a parking spot [2,3,4], adding up to 17 h a year just looking for a parking space. This obviously increases both the chance of being involved in an accident and the carbon footprint. Although these might be the most foreseeable effects, several studies also report negative effects for the driver as well. As claimed by Ponnambalam et al. [3], searching for parking requires a higher workload than regular driving. When parking availability is low, the driver’s attention is divided between two tasks: driving and searching for parking. This induces higher driver fatigue levels. Furthermore, it causes a negative feeling of delay to their final destination, leading the driver to become stressed and frustrated. Even more, if no parking spot is found after a certain amount of time, the driver might end up parking in illegal places, increasing the risk of accidents even more [5,6].

Automatic systems for real-time monitoring and reservation of parking spaces can be a turning point in parking space management in urban environments. Several methods can be found in the literature addressing real-time monitoring of parking spaces, which can be categorized into two different groups: visual and non-visual methods [2].

Non-visual methods were some of the first systems proposed for parking occupancy detection. They can be divided into global and local occupancy approaches. Global occupancy approaches count the number of vehicles entering and leaving a parking lot. For indoor parking lots, gate-arms counters are used to count the number of vehicles going through both the entrance and the exit. Furthermore, for outdoor parking lots, induction loop detectors are preferable to count the in-and-out vehicle flow. The main advantages of these systems are their low price and their fast deployment. An example of these methods can be found in [7]. However, previous works do not provide the location of the vacant spots, only the global occupancy information [8]. Local occupancy approaches provide individual information on the occupancy of every parking spot, and therefore its precise location. They typically use a detection sensor per parking space, such as ultrasonic or magnetic sensors, and LED lights as indicators of occupancy. In these cases, one sensor per spot is typically used. Along with the sensor, LED lights are commonly used as well. An example can be found in [9]. Whereas these kinds of non-visual methods are usually the most common in closed parking lots, they are not easy to scale because of the requirement of one sensor per parking space. This is even more costly for outdoor parking slots, where additional infrastructure is required for the deployment of sensors and indicators, as there is no ceiling to set the sensors. Other types of non-visual methods are those based on crowd knowledge [2]. These methods use human interaction to collect information on the parking occupancy status. Shi et al. [10] present an example of these methods. In it, partial information about vacant parking spots is collected via the interaction of drivers (users) with an app that includes a map and geolocalization capabilities. Then, the missing information is inferred using a logistic regression method. However, these methods can miss vacant parking spots due to a lack of collected data. In addition, even when enough data has been retrieved, the reliability of the crowd-collected knowledge is not guaranteed.

Visual methods are especially appealing for indoor and outdoor parking lots as a single camera is able to monitor an area with over 100 parking spots. Even more, already existing surveillance cameras could be reused, reducing the cost and deployment time. These methods infer parking space locations and markings from the acquired images to analyze the occupancy. Such systems usually require advanced computer vision algorithms to process the rich image-based information [8].

There are typically two stages involved: parking slot detection and occupancy estimation. The first stage determines the location of parking slot candidates on the parking place image, typically when they are empty. In the Chen et al. proposal [11], the marking corners of parking spaces are estimated using the FAST algorithm to predict every parking slot. Later, detected corners are filtered and used to estimate possible parking lines via the RANSAC method. Finally, parking slots are inferred by grouping parallel lines. In the Xiang et al. design [12], Haar-like image features were extracted and delivered to a cascade of Gentle AdaBoost classifiers to determine possible parking slots. Varghese et al. [13] use a background subtraction algorithm to determine potential parking slots for non-delimited spaces (no markings are available) by monitoring temporal changes of parked cars.

The second stage, parking occupancy, estimates the free parking spaces assuming that the location of the parking slots on the image is already known, either using one of the previous parking slot detection algorithms or manually delimiting the existing spaces. In the Chen et al. proposal [11], a region-growing algorithm and a Canny edge detector for feature extraction are applied inside every parking slot, and the resulting image features are delivered to a Naive Bayes classifier that predicts the occupancy. Varghese et al. [13] suggest a Bag of Features method followed by a Support Vector Machine (SVM) classifier for the occupancy estimation. Almeida et al. [14] proposed a Local Phase Quantization (LPQ) for feature extraction followed by an SVM classifier. In the Vítek et al. method [15], a Histogram of Oriented Gradients (HOG) was used to obtain the feature descriptor, followed by an SVM classifier. These previous methods use shallow learning techniques for inference; however, a significant improvement can be achieved using deep learning ones, such as the ones presented next, as the relevant features are automatically extracted from images. In the Xiang et al. design [12], a deep learning model based on VGG-16 [16] architecture is used to perform the per-slot occupancy prediction. Another Convolutional Neural Network (CNN) architecture based on a reduced version of AlexNet is proposed by Amato et al. [17] to estimate the occupancy of every individual parking spot. Similarly, Farley et al. [18] propose both LeNet and AlexNet architectures to identify parking occupancy in real time. In the Rahman et al. proposal [19], a low complex architecture called EffecientParkingNet, is developed to estimate the parking availability under hardware computational limitations. In the Ellis et al. design [20], a multi-layer perceptron style architecture, called CoarseNet, along with two reduced versions of DenseNet, are proposed as lightweight architectures for parking occupancy detection. In [21], Nyambal et al. implemented a LeNet network with Nesterov’s Accelerated Gradient solver. Šćekić et al. [22] propose a CNN-based algorithm called Faster R-CNN. Similarly, Muhammad et al. [23] propose a ResNet50 algorithm for parking occupancy detection. In the Rafique et al. study [24], in order to manage parking availability, YOLOv5 is proposed for vehicle detection instead of parking slot classification. All the previous occupancy estimation studies work at the local level, making predictions based on the local image information delimited by single parking spots. Therefore, they largely depend on the quality of the available information about the location of the parking spaces. There are two main drawbacks to this type of architecture. First, no context information is used to predict the occupancy of each parking space, which might create highly ambiguous and error-prone circumstances due to illumination changes, background occlusions, inter-vehicle occlusions, and effects of the camera perspective. A very likely challenging scenario is given by camera perspective. As a single camera is in charge of several slots, lens distortions are usually encountered. Another challenge leading to ambiguous situations is given by background occlusions. Objects such as trees, streetlights, or traffic lights can be found in large parking spaces, partially blocking the view of several parking spaces. In addition, as the system is expected to work 24 h a day, dramatic illumination changes are also frequent due to over or underexposure of the camera sensor. Figure 1 shows some examples of these ambiguous situations. Second, highly precise annotations of the regions delimiting every parking slot must be provided, which can be challenging due to occlusions and camera perspective, and very especially if typical bounding-box-based annotations are used, particularly when using simple bounding boxes (a common strategy to decrease the annotation computational cost), since it is not possible to accurately enclose them.

To partially alleviate the lack of contextual information, the information of three consecutive space regions is used by Vu et al. [25] to obtain the middle slot status. This work uses a Siamese CNN architecture for extracting feature descriptors. Then, a convolutional Spatial Transformer Network is proposed to compensate for the geometrical distortion of individual parking spots due to the perspective, and finally, another CNN architecture is used to determine the occupancy. However, more contextual information would be beneficial, as well as less exigent quality annotations.

This work proposes a new approach to detecting free parking spots that solves, or at least largely alleviates, the previous problems by using all the available contextual image information to predict the occupancy of every parking space, and by relaxing the quality of the location information of every parking spot. This system has two main advantages: (a) to use the whole image’s contextual information that state-of-the-art architectures lack, and (b) to allow more flexible annotation requirements for parking slot locations that reduce its dependency on the prediction results, as the proposed system does not need to know the location, just the occupancy. For this purpose, a multi-branch output neural network (MBONN) is proposed, which is composed of a backbone and a set of fully connected (FC) based branches. The backbone extracts a common feature map from the input image, which is then independently processed by each output branch, one per parking slot, to determine the occupancy status of every one of them. Thus, all the contextual information is considered and differently processed to infer the occupancy, addressing the challenges that involve each parking slot (different illumination, occlusions, and geometric distortions due to perspective). Additionally, the backbone is based on the high efficient new ConvNeXt architecture. On the other hand, the parking slot annotations and their quality are relaxed, just requiring a point-based annotation indicating the approximate center of every parking slot instead of complex polygons or oriented bounding boxes. The key is that the proposed system uses all the contextual image information to predict the occupancy of every slot instead of the strict area of every parking space, and therefore is not critically dependent on the accurate delimitation of them. To obtain precise conclusions about the system’s performance, it will be evaluated with several public datasets, including a new proposed dataset called ETSIT Parking Lot Occupancy Database (ETSIT) [26]. The system will allow people to spend less time driving and reduce both pollution and traffic, while also decreasing the probability of having a traffic accident.

The main contributions of this system are as follows:A new architecture is proposed, composed by multiple neural network output branches where each branch utilizes all available contextual image information to specifically analyze and predict a single parking slot. In this architecture, each branch is responsible for adapting the system to the unique needs and characteristics (perspective, distortions, etc.) of its assigned slot. By analyzing all available information in a personalized manner for each parking slot, each branch is better equipped to handle challenging scenarios.Each parking slot is simultaneously predicted in parallel by a dedicated branch in the MBONN. This allows all slots to be predicted at the same time, in contrast to state-of-the-art approaches that predict serially one slot at a time. Consequently, the proposed MBONN significantly reduces computational complexity.The annotation process is simplified by the proposed system, which does not require exhaustive boundary information to locate each parking slot, unlike other state-of-the-art approaches. For ground-truth annotations, only the occupancy status is necessary. However, although the system does not receive explicit information about the order of the parking slots during training, it implicitly learns to locate them because the occupancy information is consistently provided in the same order. Specifically, since the annotator always follows a predetermined order when annotating the slots, the system learns to associate each occupancy status with the corresponding parking slot during training, without being explicitly informed about the order.

The structure of this document goes as follows. Section 2 describes the proposed parking lot recognition system. Section 3 details the used metrics and gives the obtained results, while also making comparisons with other state-of-the-art works. Finally, conclusions and future work are drawn in Section 4.

## 2. Applied Methodology

In this project, a free parking space recognition system using surveillance cameras is presented. For this purpose, a multi-branch output neural network architecture was developed, as can be seen in Figure 2. Inputs are color images acquired from the parking scene. These images are processed by a backbone that is responsible for adaptively computing a high-discriminative and global feature map that semantically represents the relevant image information (parking slots, cars, occluding objects, and so on). That common feature map is independently processed by a set of FC network branches, the multi-branch output, where each one uses all the image information (the global feature map computed by the backbone) to estimate if a specific parking slot is free.

The proposed system can work with a number of CNN-type architectures, including ResNet [27], Xception [28], NASNet [29], and ConvNeXt [30]. The system performance using the previously mentioned backbones has been studied, all of them with an input image of size 267 × 200. All of the different backbones were able to give great performances. However, the ConvNeXt backbone was chosen for the final implementation. The ConvNeXt network is the newest one, and it is built following the design ideas of MetaFormer [31], which combines the high-level design of Transformers while substituting the Self-Attention module with a 3D convolutional block that has a lower computational cost.

### 2.1. ConvNeXt-Based Backbone

ConvNeXt is one of the latest CNN architectures that evolves the ResNet architecture by adopting the design principles of Visual Transformers [32], achieving similar performance to that of the former at a significantly lower computational cost. The ConvNeXt architecture, which can be seen in Figure 3, can be divided into three hierarchical entities called stages, blocks, and layers. Following this division, its structure contains five stages, each of which is composed of a fixed number of blocks. Similarly, each block is composed of a fixed number of layers.

The first stage, in purple, also known as the stem cell, is the network’s input and consists of a single block with two layers. The first layer processes the input image with convolutions using 64 kernels of size 4 × 4 and stride of 4 × 4, in such a way that there is no pixel overlapping between consecutive applications of a filter. This first layer is equivalent to the operation of tokenization and feature embedding of the Visual Transformer architecture. The second layer is a Layer Normalization (LN) [33] to avoid the problem of vanishing and exploding gradients [34].

The next four stages are composed of a concatenation of 4, 4, 10, and 3 blocks, of which the first 3, 3, 9, and 3 blocks, respectively, contain five layers each, where the input size (W) of the first stage is 96, and for each stage, it gets doubled. The first two layers are a 7 × 7 depthwise convolution with W kernels and an LN, which combine the information among tokens (token mixer) [31]. The other three layers are a 1 × 1 convolution, a Gaussian Error Linear Unit (GELU) [35], and a 1 × 1 convolution, forming a sub-block called channel mixer. This sub-block combines the information across channels, following an inverted bottleneck design [30], in which the hidden dimension is four times wider than the input and output of the sub-block. The architecture design was presented by MobileNetV2 [36], and since then it has gained popularity in several advanced architectures [37,38].

The last block of the first three stages is composed of a concatenation of two layers: an LN and a 2 × 2 convolution with a stride of 2 × 2.

### 2.2. Multi-Branch Output

The network output is formed by a set of FC branches that make independent predictions for each parking slot from the common feature map computed by the backbone. The number of output branches is equal to the number of parking slots in each scene since each branch processes independently and differently the common feature map provided by the backbone to predict the occupancy of each of them. This allows to consider specific challenges per parking space, such as background occlusions, intra-car occlusions, and perspective distortions. Each branch has been implemented using a block of independent FC layers, ReLU activation layers, and drop-out layers. The last layer is a sigmoid activation function that provides occupancy probabilities for every parking slot, which are finally thresholded to obtain a hard decision about the occupancy status.

### 2.3. Network Training

The proposed system has been evaluated on an Intel Core i9-7900X @ 3.30 GHz CPU along with an Nvidia Titan Xp (12 GB) GPU. TensorFlow using CUDA in Python was used to implement the model, using PyCharm as the computing platform.

For study purposes, each dataset has been divided into three random subsets following an 80%, 10%, 10% ratio. The larger subset has been used for training purposes, whereas the other two subsets have been used for validation and testing, respectively. The binary cross-entropy cost function, given by Equation (Equation 1), where *N* is the number of parking slots, was used to compare the predictions made by the network with the ground-truth values. With the aim of minimizing the cost function ($H_{p}\left(p\right)$) for each epoch, training is guided by an optimization procedure based on gradient descent algorithm [39] that minimizes the previous cost function using an iterative framework.
(1)$$H_{p}\left(p\right)=-\frac{1}{N}\sum_{i=1}^{N}y_{i}\times \log\left(p\left(y_{i}\right)\right)+\left(1-y_{i}\right)\times \log\left(1-p\left(y_{i}\right)\right)$$

Adam optimization has been adopted. This algorithm can be expressed with Equations ([Disp-formula FD2a-sensors-23-03329]) to ([Disp-formula FD2f-sensors-23-03329]):
(2a)$$g_{t}=\nabla_{\theta }f_{t}\left(\theta_{t-1}\right)$$
(2b)$$m_{t}=\beta_{1}\times m_{t-1}+\left(1-\beta_{1}\right)\times g_{t}$$
(2c)$$v_{t}=\beta_{2}\times v_{t-1}+\left(1+\beta_{2}\right)\times g_{t}^{2}$$
(2d)$$\hat{m_{t}}=\frac{m_{t}}{\left(1-\beta_{1}^{t}\right)}$$
(2e)$$\hat{v_{t}}=\frac{v_{t}}{\left(1-\beta_{2}^{t}\right)}$$
(2f)$$\theta_{t}=\theta_{t-1}-\frac{\alpha \times \hat{m_{t}}}{\sqrt{\hat{v_{t}}}+\varepsilon }$$
where $f_{t}\left(\theta \right)$ is a stochastic scalar function that is differentiable with reference to parameters θ; $m_{t}$ and $v_{t}$ represent the first-order momentum and second-order momentum, respectively; $\beta_{1}$ and $\beta_{2}$ are their decay rates; $\hat{m_{t}}$ and $\hat{v_{t}}$ are the corrected estimators of $m_{t}$ and $v_{t}$; α represents the step-size; and ε is a small constant used to prevent the denominator from becoming zero.

The hyper-parameter settings that have been used to train the network, such as the learning rate or the total number of epochs, are listed in Table 1.

### 2.4. Metrics

Accuracy of the empty parking space prediction has been the metric selected to evaluate the system’s performance, as it was the one used in other proposals. It is defined as the percentage of correct predictions; this is, the percentage of correctly detected free parking slots in every image and across all images. Equation (Equation 3) gives its mathematical expression:(3)$$\mathrm{Accuracy}=\frac{\mathrm{Number}\;\mathrm{of}\;\mathrm{correct}\;\mathrm{predictions}}{\mathrm{Total}\;\mathrm{number}\;\mathrm{of}\;\mathrm{predictions}}$$

Both enumerator and denominator can be expressed using the metrics True Positives (TP), True Negatives (TN), False Positives (FP), and False Negatives (FN), resulting in a new expression of the accuracy given by Equation (Equation 4):(4)$$\mathrm{Accuracy}=\frac{\mathrm{TP}\;+\;\mathrm{TN}}{\mathrm{TP}\;+\;\mathrm{TN}\;+\;\mathrm{FP}\;+\;\mathrm{FN}}$$

TP is the number of parking slots that have been correctly identified as free. Similarly, TN indicates the number of parking slots that have been correctly identified as occupied. On the other hand, FP is the number of occupied parking slots that have been predicted as free. Finally, FN is the number of free parking slots that have been predicted as occupied.

## 3. Results

First, the datasets used to evaluate the proposed system are presented. The Section 3.2 shows the quantitative results. It provides the performance of the system with a comparison of the different backbones and with other state-of-the-art solutions, as well as its sensitivity. Finally, some qualitative results are provided in the Section 3.3.

### 3.1. Datasets

Four different databases have been used to evaluate the proposed system for detecting the occupancy of parking slots: ETSIT [26], PUCPR [14], UFPR04 [14], and UFPR05 [14]. The ETSIT database contains day and night images of a complex parking scenario located at the ETSI Telecomunicación of the Universidad Politécnica de Madrid, and it presents important challenges such as perspective distortions, background occlusions, and different illumination conditions. Some image examples were shown in Figure 1. This dataset was chosen because of its extreme conditions of the farthest slots, which suffer from both a high degree of occlusion and a reduced region size due to the camera perspective. The scene contains 21 annotated parking slots (see Figure 4a). The other three datasets are subsets of the PKLot database [14], which is the most extensive database in the state of the art for parking spot detection. The PUCPR subset contains images of a parking lot located at the Pontifical Catholic University of Parana, which contains 100 parking spaces (see Figure 4b). The other two subsets, UFPR04 and UFPR05, have been acquired at the Federal University of Parana. The UFPR04 dataset contains 28 parking slots (see Figure 4c), while the UFPR05 has 40 parking spaces (see Figure 4d). In all three cases, the datasets include various weather, occupancy, and luminosity circumstances.

Each dataset is composed of a set of images and its corresponding ground-truth file. The latter is a text file containing the occupancy of each parking slot encoded as a binary vector. A value of 1 means that the spot is empty, while 0 means it is occupied. All datasets are divided into three subsets: training, validation, and test. General information about each database, such as the number of images or parking spots, can be found in Table 2.

Figure 4a–d show the numbered slots of the ETSIT, PUCPR, UFPR04, and UFPR05 databases, respectively. Notice that these datasets do not consider all of the parking slots in the different scenarios.

As can be observed in Figure 4, the different slots have been numbered for visualization purposes. However, no location is needed for the system. For this method, the occupancy status is enough. The occupancy information has to be given in the same order for every image. This way, the system learns to locate the parking spots on its own.

### 3.2. Quantitative Results

The proposed system has been evaluated using the four previous databases (ETSIT, PUCPR, UFPR04, and UFPR05) and the accuracy metrics. First, a comparison using different backbones (Resnet50, Xception, NASNetLarge, ConvNeXtBase, and ConvNeXtTiny) is presented. Then, a comparison with other works in the state of the art is included.

#### 3.2.1. Comparison with Different Backbones

Table 3 shows the obtained accuracy results per database while Table 4 shows its standard deviation, comparing different backbones for the first stage of the proposed multi-branch output neural network architecture. The input image size of Resnet50, Xception, NASNetLarge, ConvNextBase, and ConvNextTiny is 267 × 200, quite lower than the original image resolution due to the memory restrictions to train the neural networks models, which is a typical situation in deep neural networks oriented to image processing applications. This can be problematic for the farthest parking spaces and also for those that suffer from background occlusions (due to trees, streetlights, etc.).

Table 3 shows the average accuracy for each dataset for the different backbones using input images of the reduced resolution of 267 × 200. The outcomes for all databases using any of the backbones are extremely satisfactory. For every scenario, the occupancy of parking spaces was predicted with an accuracy greater than 99.1%. This fact proves that the proposed multi-branch network design can effectively infer the occupancy of parking slots, independently of the chosen backbone.

Similarly, Table 4 shows the standard deviation of said accuracy for the different backbones. As can be observed, the ETSIT dataset has the lowest values for all the backbones. Since this dataset is the largest in terms of the number of images, the system is able to gain better knowledge of more complex scenarios, and therefore its performance is more robust. Regarding the different backbones, it can be seen that the ConvNeXt architecture is the one to yield a lower standard deviation.

From the results of Table 3 and Table 4, it can be concluded that the model operates as intended, given the high accuracy outcomes and its small standard deviation. Therefore, the system is consistent and stable through all parking slots, despite the fact that the prediction conditions of the farthest lots (see parking slots 6, 18, 19, and 20 in Figure 4a) are much more difficult than the closest ones. On the other hand, the system proves to be robust to 24 h operation (daytime and nighttime), considering that 30% of the images have been acquired in the nighttime. However, a slightly better performance is obtained form the ConvNeXt architecture, as its average accuracy values are higher while the standard deviation is lower.

Even if the ConvNeXt backbone has greater performance accuracy, all backbones are able to obtain great values. Therefore, all backbones are suitable for the system. The total number of parameters of the entire model has also been considered in order to determine which architecture would be most convenient for low-cost and embedding processing hardware. They are portrayed in Table 5. As can be observed, and attending to the memory requirements of the embedded hardware, the most suitable architecture is the ConvNeXtTiny algorithm, as it has way less number of parameters.

The proposed system is computationally efficient and well-suited for real-time applications since it is capable of processing an average of 40 frames per second, which is more than enough for most use cases. For parking management and monitoring, even predicting just one frame per second is enough because the status of a single parking slot typically remains unchanged for a while when a car is parked. Therefore, the system can successfully meet the requirements of such applications.

#### 3.2.2. Sensitivity Analysis of the System

The proposed MBONN system was subjected to a sensitivity analysis to evaluate its performance under varying levels of distortion and illumination conditions. Table 6 presents the accuracy results obtained for each parking spot in the ETSIT database at different distortion levels. The table reveals that the performance of the system remained stable across all the parking spots, regardless of the level of distortion. This suggests that the system is robust and reliable even when the images are degraded or distorted. The performance of the proposed system under varying illumination conditions was also evaluated. Table 7 shows the average accuracy results obtained for both day and night scenarios in the ETSIT database. The test images were divided into two categories based on their light levels, and each scenario was evaluated separately. The results reveal that the performance of the system remained stable and reliable in both scenarios, indicating that it is robust even under low lighting conditions.

#### 3.2.3. Comparison with State-of-the-Art Works

Unlike state-of-the-art architectures, the proposed system needs no location annotation. Therefore, it is expected to be more robust to challenging conditions, as the system is the one that learns how to locate the spatial position of the different spots. To confirm this statement, a comparison with several state-of-the-art works has also been obtained through a set of experiments. The first one evaluated the accuracy for each system with the PCUPR, UFPR04, and UFPR05 datasets, as well as the average accuracy performance on the PKLot database. For each dataset, 45% of the images have been used for training, 5% have been used for validation, and the other 50% have been used for testing. Table 8 shows a detailed comparison of the proposed multi-branch system to numerous current state-of-the-art works with this configuration. The accuracy ratings obtained by each system on a variety of datasets, proposed by [14], are used to make the comparison. Table 8 shows that the suggested multi-branch system performs exceptionally well in terms of accuracy on all datasets. This outstanding performance can be ascribed to the suggested ability of the multi-branch architecture to appropriately handle several views and a wide variety of slots.

Table 8 shows that five of the algorithms that were compared achieved accuracies over 99.1%. While these results are impressive, it is important to note that the information provided in the state-of-the-art studies may not be sufficiently exhaustive for fully reproducing the experiments. Therefore, it is difficult to definitively say which algorithm is the best, since to ensure that the experiments are conducted under equal conditions, a 1% error rate is typically used as a benchmark. Consequently, any of the five systems with an accuracy above 99% can be considered to perform similarly.

Computational time is another important factor to consider, but unfortunately, this information is not always provided in the state-of-the-art literature. However, based on our analysis, we found that many existing systems only predict one parking slot at a time, i.e., making predictions in series, which can be a time-consuming process. In contrast, the proposed MBONN architecture is designed to predict multiple parking slots simultaneously, in parallel, which should improve computational time compared to traditional approaches.

As the previous datasets do not really consider challenging scenarios, the second experiment consisted of evaluating the proposed system and the mAlexNet [17] implementation, which yielded one of the highest performances, in a more challenging scenario. For this purpose, the ETSIT dataset has been used with a training, validation, and testing ratio of 80%, 10%, and 10%, respectively. The obtained results for both methods can be seen in Table 9. In this case, both systems give great performances; however, the multi-branch output system has been able to perform slightly better. To prove that the system is more robust to challenging scenarios, the errors made by both systems have been collected. Upon examination, it has been discovered that the mAlexNet implementation is more prone to fail when partial occlusions are encountered. Figure 5 depicts an image of this kind of scenario. Furthermore, it has also been discovered that the proposed method is able to deal better with perspective distortions. As no spatial information is provided to the system, it is able to correctly predict the output even if the camera angle barely allows its view. Figure 6 shows an example of such distortion.

Finally, the convergence rate of the ConvNextTiny system and the mAlexNet system [17] has also been analyzed. To do so, two trials with a reduced number of training images for the ETSIT dataset have been computed. The original division of this experiment has been 45% training, 5% validation, and 50% testing. For the first trial, just 30% of the training images have been used, while for the second trial, the number has been increased to 50%. As it can be observed in Table 10, for 30% of the training images both systems give similar performances. However, when 50% of the images are used to train, the proposed method is able to achieve higher accuracy values. Therefore, the proposed system needs a lower number of images to yield accuracies over 99%.

### 3.3. Qualitative Results

As was previously stated, the system is able to correctly perform under different lighting and occupancy conditions. Some examples of the obtained results on critical images using the ConvNextTiny backbone are shown in Figure 7. An interesting scenario can also be seen in Figure 8. In this case, an error due to a temporal full occlusion is portrayed. A truck is driving through the parking lot, causing occlusions along the way. This is arguably a true error since the spot is not visible in the image, and once the truck is gone, the system will recover its regular performance.

## 4. Discussion

In this study, an automatic occupancy detection system for outdoor parking lots has been presented based on a multi-branch output neural network architecture. The main contributions are to use the whole image contextual information that state-of-the-art architectures are lacking and to allow more flexible notation requirements for parking slot locations that reduce its dependency on the prediction results.

This work has studied several backbones for the proposed systems. All of them (Resnet50, Xception, NASNetLarge, ConvNextBase, and ConvNextTiny) with an input image size of 267 × 200, which is lower than the original image resolution due to the memory restrictions to train the neural networks models. Regarding accuracy, it has been concluded that any of them are suitable for this task. However, both versions of the ConvNeXt have performed slightly better. On the other hand, the total number of parameters, which has been the deciding factor, indicates that the Tiny version of the ConvNeXt algorithm takes up the least amount of memory. Therefore, it is better suited for embedding in hardware with limited computational capacity for more cost-effective system deployment.

In addition, the proposed MBONN system has been subjected to two types of sensitivity analysis to evaluate its performance under varying levels of distortion and illumination conditions. In the first analysis, the system has been tested under different levels of distortion using the ETSIT database. The results show that the performance of the system remains stable across all parking spots, regardless of the level of distortion. This indicates that the system is robust and reliable even when the images are degraded or distorted. In the second analysis, the performance of the system has been evaluated under different illumination conditions (day and night scenarios) using the same database. The results reveal that the accuracy of the system remains stable and reliable in both scenarios, indicating that it is robust even under low lighting conditions. Overall, these findings suggest that the proposed MBONN system is capable of accurately detecting parking slot occupancy in real time, and can perform well under challenging conditions. These results can have important implications for the development of smart parking management and monitoring systems, which require accurate and reliable detection of parking occupancy status.

On the other hand, the accuracy of the system has been compared with several state-of-the-art works using a set of experiments on various datasets, and the results have shown that the proposed system performs exceptionally well in terms of accuracy on all datasets. The proposed system has been designed to predict multiple parking slots simultaneously, which should improve computational time compared to traditional approaches. The robustness of the system has been demonstrated through an experiment using a more challenging dataset, where it has outperformed one of the highest-performing state-of-the-art methods. The proposed method has also been found to be able to handle partial occlusions and perspective distortions better than other methods. Finally, the proposed system has been found to achieve higher accuracy values with a lower number of training images than the compared state-of-the-art works.

Improving the available datasets should be considered for future work lines. To do so, two major objectives must be met. The first is to broaden the range of available datasets. Even if excellent results have been obtained, some of the observed prediction errors may be corrected with additional training data. The second approach is to work with pre-existing data. The most occluded slots in the available datasets have not yet been identified. This means that performance in more difficult scenarios has not been evaluated. As a result, improving their notation is critical to creating a more robust system.

Another possible future line of work would be to extend the analysis to the time domain in order to obtain usage and temporal prediction statistics. The main idea behind this future path would be to improve the proposed system through data collection and analysis. Investigating the application of transfer learning techniques in this scenario could also be a viable future strategy. As a result, the system can be adapted to new scenarios with minimal additional changes. Finally, another possible future project would be the system’s adaptation for embedded hardware.

## Figures and Tables

**Figure 1 sensors-23-03329-f001:**
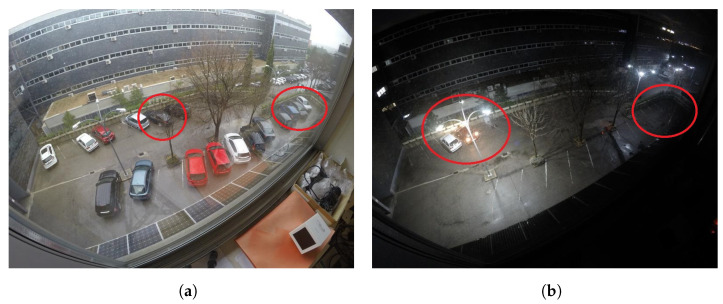
Examples of challenging scenarios given by illumination changes, background occlusions, inter-vehicle occlusions, and effects of the camera perspective (such as lens distortion). (**a**) Example of background occlusion and lens distortion; (**b**) example of illumination conditions.

**Figure 2 sensors-23-03329-f002:**
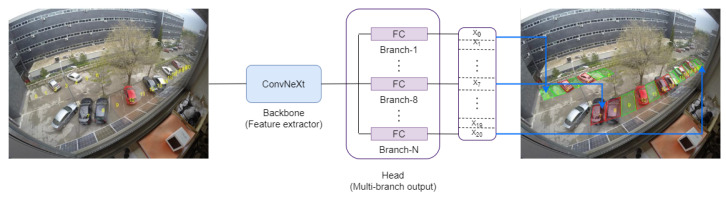
Input–Output system architecture. The backbone processes the color images acquired from the parking scene to compute a global feature map. The multi-branch output independently processes the common feature map, so each branch uses all the image information to estimate if a specific parking slot is free.

**Figure 3 sensors-23-03329-f003:**
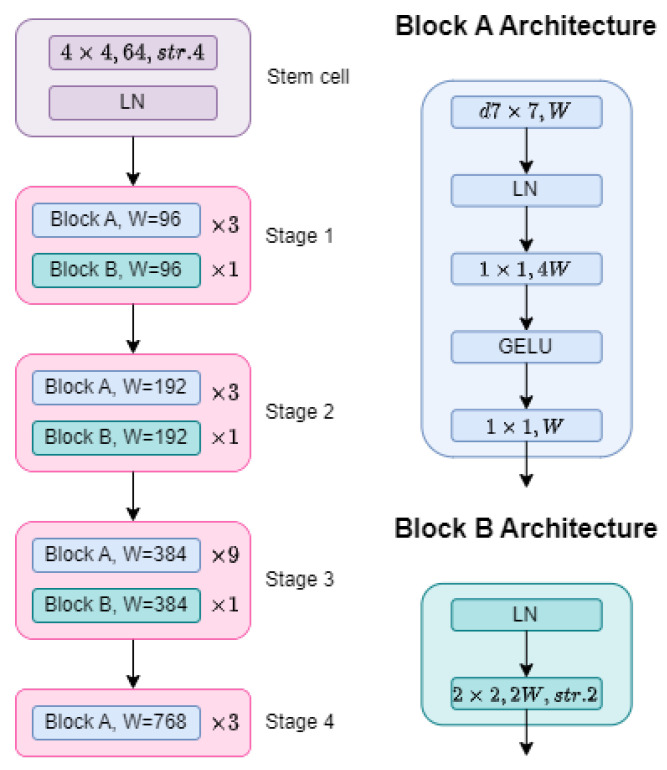
ConvNeXt architecture. The left part of the figure shows the general connection of the three hierarchical entities: stages, blocks, and layers. The right part of the figure represents the composition of a single block into layers.

**Figure 4 sensors-23-03329-f004:**
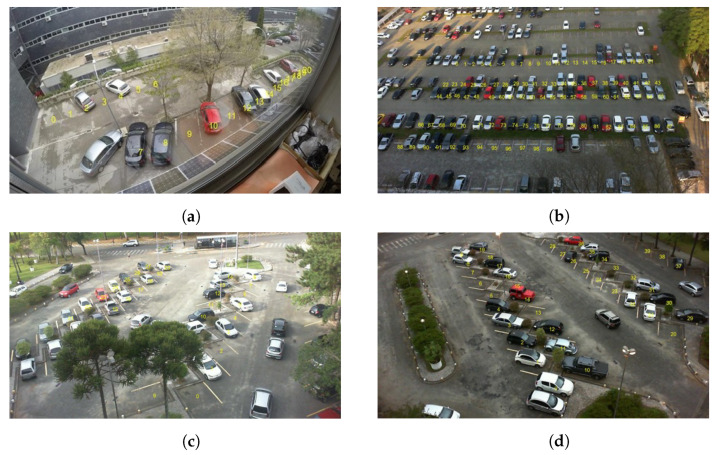
Example images of the four employed datasets, with all the considered slots numbered from 0 to N-1. (**a**) a sample of the ETSIT database; (**b**) a sample of the PUCPR database; (**c**) a sample of the UFPR04 database; and (**d**) a sample of the UFPR05 database.

**Figure 5 sensors-23-03329-f005:**
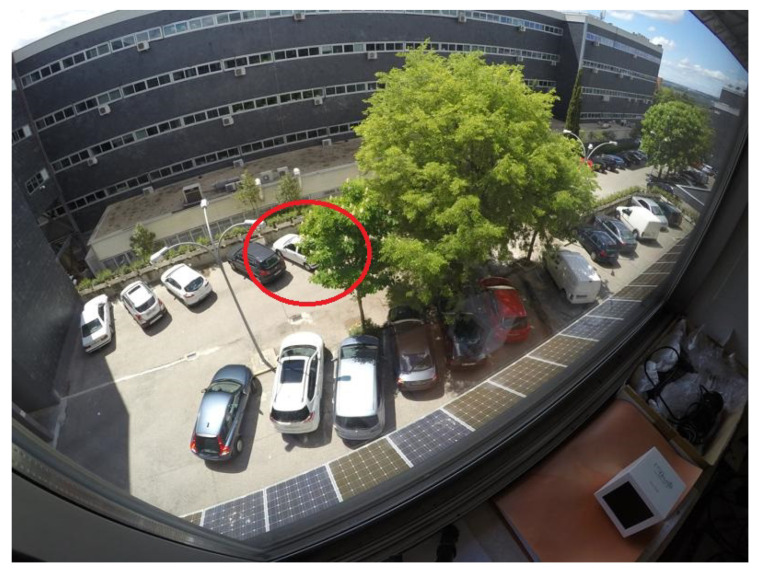
Example of a parking scenario of the ETSIT dataset with a partial occlusion given by a tree.

**Figure 6 sensors-23-03329-f006:**
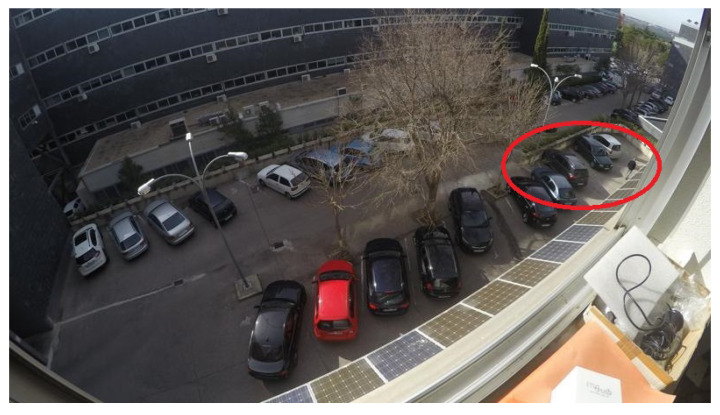
Example of a parking scenario of the ETSIT dataset with a lens distortion given by the camera perspective.

**Figure 7 sensors-23-03329-f007:**
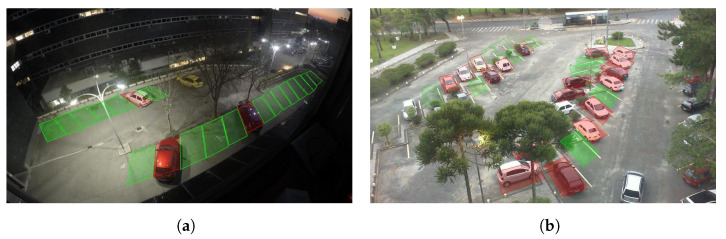
A sample of predictions obtained with the proposed multi-branch output system for two different datasets with different illumination and occupancy conditions. (**a**) Example of prediction for the ETSIT database with low lightning conditions; (**b**) example of prediction for the UFPR04 database with medium occupancy.

**Figure 8 sensors-23-03329-f008:**
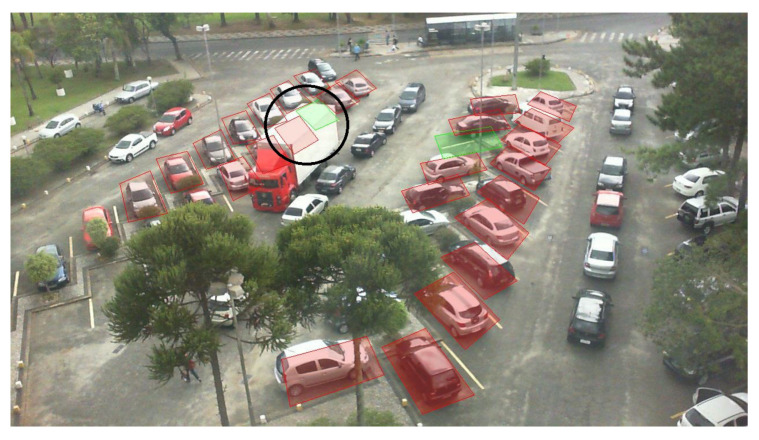
Example of an incorrect prediction given by the proposed system testing with the UFPR04 database due to a temporal full occlusion between vehicles.

**Table 1 sensors-23-03329-t001:** Hyper-parameters used to train the network.

Hyper-Parameter	Value
Learning rate	$10^{-4}$
Number of epochs	50
Batch size	16
Image size	267 × 200
Weight decay	$10^{-6}$

**Table 2 sensors-23-03329-t002:** Brief description of the four employed datasets, indicating the total number of images available and the number of parking slots considered in each scenario.

Database	Total Images	Number of Parking Slots
ETSIT	31,611	21
PUCPR	4474	100
UFPR04	5104	28
UFPR05	4153	40

**Table 3 sensors-23-03329-t003:** Average accuracy of the proposed system for each of the tested backbones, computed with all the employed databases. Each column represents the accuracy performance given by one of the backbones, while each row indicates the accuracy obtained in a particular dataset.

	Resnet50	Xception	NASNet (Large)	ConvNeXt (Base)	ConvNeXt (Tiny)
ETSIT	99.98	99.94	99.96	99.97	99.99
PUCPR	98.87	98.20	98.41	99.73	99.12
UFPR04	99.95	99.87	99.92	99.94	99.95
UFPR05	99.77	98.73	99.14	99.84	99.83

**Table 4 sensors-23-03329-t004:** Standard deviation of the performance accuracy through all the slots for each backbone.

	Resnet50	Xception	NASNet (Large)	ConvNeXt (Base)	ConvNeXt (Tiny)
ETSIT	0.0180	0.0990	0.0371	0.0225	0.0090
PUCPR	0.6000	0.9032	0.7612	0.2679	0.4751
UFPR04	0.0771	0.1335	0.1073	0.1103	0.0760
UFPR05	0.3098	0.7007	0.6074	0.2857	0.2177

**Table 5 sensors-23-03329-t005:** Number of parameters of the complete model for each backbone and each dataset.

	ETSIT	PUCPR	UFPR04	UFPR05
Resnet50	155,730,837	155,811,812	155,738,012	155,750,312
Xception	153,004,605	153,085,580	153,011,780	153,024,080
NASNetLarge	345,051,751	345,132,726	345,058,926	345,071,226
ConvNeXtBase	137,920,661	138,001,636	137,927,836	137,940,136
ConvNeXtTiny	65,591,413	65,672,388	65,598,588	65,610,888

**Table 6 sensors-23-03329-t006:** Sensitivity analysis of the image distortion for the ETSIT database and the ConvNeXtTiny backbone. Accuracy values per parking spot have been computed to portray the stability of the proposed system.

Spot	Accuracy	Distortion Level
0	100.00	Low
1	100.00	Low
2	100.00	Low
3	100.00	Low
4	100.00	Low
5	99.97	Medium
6	99.97	Medium
7	100.00	Low
8	100.00	Low
9	100.00	Low
10	100.00	Low
11	100.00	Medium
12	100.00	Medium
13	100.00	Medium
14	100.00	Medium
15	100.00	High
16	100.00	High
17	100.00	High
18	100.00	High
18	100.00	High
20	100.00	High

**Table 7 sensors-23-03329-t007:** Sensitivity analysis of the illumination of the image for the ETSIT database and the ConvNeXtTiny backbone. Accuracy values for day and night scenarios have been computed to portray the stability of the proposed system.

Scenario	Accuracy	Light Level
Day	100.00	High
Light	99.99	Low

**Table 8 sensors-23-03329-t008:** Comparison of accuracy values with state-of-the-art works for the PCUPR, UFPR04, and UFPR05 datasets. Additionally, the average accuracy for the PKLot database has also been compared.

	PUCPR	UFPR04	UFPR05	PKLot
LQP + SVM [14]	99.58%	99.55%	98.90%	99.34%
HOG + SVM [15]	94.00%	96.00%	83.00%	91.00%
Background subtraction + SVM [13]	N/A	99.72%	N/A	N/A
ResNet50 [23]	N/A	N/A	N/A	99.67%
mAlexNet [17]	99.90%	99.54%	99.49%	99.64%
YOLOv5 [24]	N/A	N/A	N/A	99.50%
LeNet [21]	N/A	N/A	N/A	93.00%
ConvNeXt (Tiny) multi-branch system	98.30%	99.73%	99.38%	99.14%

**Table 9 sensors-23-03329-t009:** Comparison between the ConvNeXtTiny multi-branch system and the mAlexNet system [17] for the ETSIT dataset.

	ETSIT
mAlexNet [17]	99.93%
ConvNeXtTiny multi-branch system	99.99%

**Table 10 sensors-23-03329-t010:** Comparison between the ConvNeXtTiny multi-branch system and the mAlexNet system [17] for the ETSIT dataset with reduced training sets.

	Training Percentage	Accuracy	Training Percentage	Accuracy
mAlexNet [17]	30%	95.05%	50%	95.92%
ConvNeXtTiny multi-branch system	30%	95.74%	50%	99.13%

## Data Availability

Not applicable.

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
