# Peer review of "Visual Parking Occupancy Detection Using Extended Contextual Image Information via a Multi-Branch Output ConvNeXt Network"

_sensors, 2023, doi:10.3390/s23063329_

Round 1

Reviewer 1 Report

This paper tackles the problem of estimating free parking slots in a parking space using computer vision. A new algorithm based on neural networks is proposed. Publicly available datasets are used for testing the effectiveness of the newly proposed approach. Good thing is that used datasets contain realistic images having problems with distortion, occlusions, and different illuminations. A comparison with the state of the art approaches is also done.

The proposed approach obtains only the occupancy status and not the location status. Thus, the problem is alleviated. For a better parking space management system, the location can also play an important role taking into account the number of free parking slots, and their user correlation (disabled persons, families with small children, etc.) to enable assigning/reservations of appropriate parking slots for entering cars. A comment about such a possible augmentation would be welcome.

You mention challenging scenarios, but no description of what a challenging scenario is can be found. It would be good to analyze how distortion, occlusions, and different illuminations affect your approach in comparison to existing ones. Thus, a sensitivity analysis in controllable conditions would be welcome commenting on the influence of individual problematic input image issues.

To improve the paper quality consider also the following:

-       Metrics description should be part of the second chapter;

-       Table 5 presents accuracy? Add this to table 5 caption;

-       Always precisely denote your referenced item (table, figure, equation);

-       Comment also on the computational and training complexity of the compared approaches.

There are additional comments in the attached PDF.

Reviewer 2 Report

This paper presents visual parking occupancy detection using extended contextual image information via a multi-branch output ConvNeXt network. In general, this work presents some interesting findings and it is technically sound. However, it has several concerns that needs to be alleviated. Here, there are some concerns of this reviewer:

1 The contributions of this paper should be further summarized and clearly demonstrated. This reviewer suggests the authors exactly mention what is new compared with existing approaches and why the proposed approach is needed to be used instead of the existing methods. This reviewer suggests the authors use bullets (3 bullets is standard) and in each bullet explain one contribution clearly.

2 The theoretical depth of this paper needs to be strengthened. This reviewer would like to suggest the authors add some equations to better describe the principle of the used deep learning model.

3 The proposed method might be sensitive to the values of its main controlling parameter. How did you tune the parameters?

4 Please specify details of the computing platform and programming language in this study.

5 The computational cost of the proposed approach is not discussed in this work. The approach must be computationally efficient to be used in practical applications.

6 Use of deep learning model for detection is a key idea in this work. To improve the literature survey, the following recent articles with the similar scenario can be cited, and discussed: https://doi.org/10.1109/TSG.2022.3204796, https://doi.org/10.1109/IT54280.2022.9743533.

Reviewer 3 Report

paper presented "Visual Parking Occupancy Detection Using Extended Contextual Image Information Via a Multi-Branch Output ConvNeXt Network" by authors is  a new system that detects vacant parking spaces in challenging situations using color imagery processed by a novel deep learning algorithm. This is based on a multi-branch output neural network that maximizes the contextual image information to infer the occupancy of every parking space. 

The method presented need to address the following queries:

1. The process need to be compared with latest SOTA with proper visualization of results.

2. Authors need to improve the result section.

3. author need to give the clear motivation and the contribution of work.

4. The discussion section need to be improved with proper visualizations of results.

Reviewer 4 Report

Review Report for the Manuscript “Visual Parking Occupancy Detection Using Extended Contextual Image Information Via a Multi-Branch Output ConvNeXt Network

Rating the Manuscript

Originality/Novelty: Is the question original and well defined? Do the results provide an advance in current knowledge?

Yes, in the manuscript the authors focus and developed a new system that detects vacant parking spaces in challenging situations using color imagery processed by a novel deep learning algorithm.

Significance: Are the results interpreted appropriately? Are they significant? Are all conclusions justified and supported by the results? Are hypotheses and speculations carefully identified as such?

Yes, the results are interpreted well.

Quality of Presentation: Is the article written in an appropriate way? Are the data and analyses presented appropriately? Are the highest standards for presentation of the results used?

Yes, the article is written well. Data representation could be improved. Specially the figure captions could be more informative.

Scientific Soundness: is the study correctly designed and technically sound? Are the analyses performed with the highest technical standards? Are the data robust enough to draw the conclusions? Are the methods, tools, software, and reagents described with sufficient details to allow another researcher to reproduce the results?

Yes, the data is robust enough to draw conclusions and the methods, tools and methods used in the data analysis are explained properly.

Overall Merit: Is there an overall benefit to publishing this work? Does the work provide an advance towards the current knowledge? Do the authors have addressed an important longstanding question with smart experiments?

Yes. This study provides an advancement to the current knowledge. 

English Level: Is the English language appropriate and understandable?

Yes, English language in the manuscript is appropriate and understandable. 

Overall Recommendation: Accept after Minor Revisions

Given below are the comments for each section of the manuscript.

Abstract

The abstract is written and summarizes the content of the manuscript.

Introduction

Introduction is well written.

Line 28: “As claimed by [3], searching for parking requires higher workload than regular driving.”

I think it’s better if the authors mention the author of the previous study or state “it’s reported that”, instead of stating “As claimed by [3]”.

Line 32: “Even more, if no parking spot is ever found, the driver might end up parking in illegal places, increasing the risk of accident even more.”

Is there any statistic or previous reports on this? If so, I think it would be great to include the information.

Line 57:” In [8], an example of these methods can be found.” 

Again, it’s better if the authors could mention the name of the author of the reference 8, instead of stating "In [8]".

And there are several other places authors could change this.

Line 72, Line 75, Line 77, Line 83, Line 85, Line 88, Line 89, Line 91, Line 95, and Line 97.

In paragraph 6, can you briefly mention the advantages and disadvantages of each method?

Materials and Methods:

Materials and methods and section is well written.

Figures and Tables

Figure captions can be improved. Authors could provide more information on the Figure captions.

“Figure 1 Caption. Examples of illumination changes, background occlusions, inter-vehicle occlusions, and effects of the camera perspective.”

Please label the two figures separately as Figure 1 A and 1B.

References:

Some of the references are more than 10 years old. It they don’t contain important information authors could replace these with new references. 

References: 4,5,7 and 10

Also, some of the references has doi and others don’t. Please have the same format for all the references.

Round 2

Reviewer 1 Report

It can be seen that the paper quality has been improved. All of my comments have been dealt with and appropriate paper augmentations have been done. The paper is now suitable for publication. Some minor typos/writing errors are present that can be corrected during preparation of the final version of the paper.

There are additional comments for correcting minor errors in the attached PDF.

Reviewer 2 Report

Thanks to the careful revision and detailed response made by the authors. All my concerns have been well addressed, and the revised manuscript has been much improved. Therefore, I think this paper deserves to be published in its current form.

Reviewer 3 Report

Authors have improved the manuscript.